# The Risk of Avascular Necrosis Following the Stabilization of Femoral Neck Fractures: A Systematic Review and Meta-Analysis

**DOI:** 10.3390/ijerph191610050

**Published:** 2022-08-15

**Authors:** Wojciech Konarski, Tomasz Poboży, Andrzej Kotela, Andrzej Śliwczyński, Ireneusz Kotela, Martyna Hordowicz, Jan Krakowiak

**Affiliations:** 1Department of Orthopaedic Surgery, Ciechanów Hospital, 06-400 Ciechanów, Poland; 2Faculty of Medicine, Collegium Medicum, Cardinal Stefan Wyszynski University in Warsaw, Woycickiego 1/3, 01-938 Warsaw, Poland; 3Social Medicine Institute, Department of Social and Preventive Medicine, Medical University of Lodz, 90-419 Lodz, Poland; 4Department of Orthopedic Surgery and Traumatology, Central Research Hospital of Ministry of Interior, Wołoska 137, 02-507 Warsaw, Poland; 5General Psychiatry Unit III, Dr Barbara Borzym’s Independent Public Regional Psychiatric Health Care Center, 26-600 Radom, Poland

**Keywords:** avascular necrosis, femoral neck fractures, systematic review

## Abstract

Background: Avascular necrosis (AVN) of the femoral head often requires surgical treatment and is often associated with femoral neck fractures. We conducted a systematic review and meta-analysis of recent research on the risk of AVN following the stabilization of fractured femoral neck with implants in PubMed. We assessed the effect of age on AVN incidence among patients aged > 50 and younger, depending on fracture type, Garden stage, Pouwels degree, Delbet stage, and age category. We followed PRISMA guidelines. Relevant studies were defined as research articles describing real-world studies reporting on the risk of AVN following primary surgical fracture stabilization with implants, published between 1 January 2011 and 22 April 2021. Fifty-two papers met the inclusion criteria, with a total of N = 5930 with surgically managed fractures. The pooled mean AVN incidence was significantly higher among patients with displaced fractures (20.7%; 95% CI: 12.8–28.5%) vs. those with undisplaced fractures (4.7%; 95% CI: 3.4–6.0%). No significant correlation was observed between AVN incidence weighted by sample size and time interval from injury to surgery (*p* = 0.843, R^2^ = 0.01). In conclusion, the risk of AVN following femoral neck fractures was generally high, especially in patients with displaced fractures. The time from injury to surgery did not correlate with AVN incidence.

## 1. Introduction

Avascular necrosis (AVN) of the femoral head involves osteonecrosis arising from altered blood supplies to the proximal femur [1]. Each year in the US alone, approximately 10,000–20,000 new cases occur [1]. AVN can arise from several causes, which may be classified as traumatic and atraumatic. [1] Femoral fractures can decrease blood flow to the femoral head, placing patients at risk of AVN [2,3]. Indeed, the fracture of the femoral neck or dislocation of the femoral head from the acetabulum is among the most common traumatic causes of AVN of the femoral head [1].

The most common symptom of AVN of the femoral head is radiating groin pain and possessing a limited range of motion with respect to the affected hip, with pain experienced during the forced internal rotation of the hip. Physical activity may aggravate the pain but it is often also present at rest [1]. The diagnosis of AVN is usually made based on both clinical presentation and appropriate imaging (X-ray, radionuclide bone scanning, and magnetic resonance imaging [MRI]) [1]. The treatment of AVN of the femoral head depends on the advancement of necrosis at the time of diagnosis, the age of the patient, the level of pain and discomfort experienced by the patient, their overall health, comorbidities, and AVN stage (pre-or post- articular surface collapse) [1,2,3,4,5]. Non-operative methods are generally employed in patients with small- or medium-sized pre-collapse lesions [1]. Operative procedures can be divided into two categories. Joint-preserving methods (i.e., core decompression, bone grafts, biological therapies, tantalum implants, and osteotomy) are suitable for patients at pre-collapse stages. In contrast, patients with more advanced AVN generally require joint reconstruction surgery (hemi-resurfacing arthroplasty and hemipolar or bipolar hip replacement) [4].

The objective of this systematic review was to summarise recent research findings on the risk of AVN after stabilizing the fractured femoral neck with screws, plates, or other types of implants.

## 2. Materials and Methods

### 2.1. Search Strategy

The review followed PRISMA [6] (Preferred Reporting Items for Systematic Reviews and Meta-Analysis) guidelines and is available as a Appendix A. PubMed was searched on 11 May 2021 to identify relevant studies published between 1 January 2011 and 22 April 2021, using a combination of Medical Subject Heading (MeSH) and free-text terms related to hip fracture (neck of femur fracture and femoral neck fracture), surgery (internal fixation/internal fixators), and outcome of interest (avascular necrosis and avascular necrosis of femur head). Bibliographic details and abstracts for all citations identified through the PubMed search were exported into EndNote version 20 to remove duplicate papers and for screening titles and abstracts.

The entire protocol of the review, including the search strategy, is available in Appendix A. The protocol was not registered with a systematic review database.

### 2.2. Study Selection Process

A single reviewer performed screening, which was checked by a second senior reviewer. Two reviewers performed data extraction from the papers, which met the inclusion criteria. The inclusion and exclusion criteria are described in Table 1.

Relevant citations were defined as original research articles describing real-world studies (excluding case reports or case series with ≤10 cases) and reporting on the risk of AVN incidence following the primary surgical stabilization of the fracture with implants (arthroplasty procedures were excluded). We did not exclude studies based on implant types or fracture reduction methods, which may depend on patient condition and surgeon’ preference and experience. Studies focused on patients with neglected fractures who underwent surgery several weeks after the causative injury were also excluded.

Of particular interest was comparing the risk of AVN between patients managed conservatively and those whose fractures were stabilized surgically. Nonetheless, studies, in which different surgical approaches were compared were also included. No comparisons between other surgical methods were planned.

### 2.3. The Rationale for the Study

We sought to assess recent research findings on the risk of AVN after stabilising the fractured femoral neck with implants such as screws, plates, or wires in the real-world setting. We focused on research questions in which the analysis of real-world data from unselected patient populations may provide particularly relevant information, such as how the real-world incidence of AVN varies by the type of fracture and by time intervals from injury to surgery.

A previous Cochrane review published in 2001 observed lower AVN incidences with the sliding hip screw compared with different cancellous screws. However, based on the overall body of evidence with regards to all relevant post-surgical outcomes assessed in that review, no overall recommendations on the choice of implant were provided [7]. We therefore did not attempt to conduct comparisons between specific implant types.

### 2.4. Dependent Variables Studied

#### 2.4.1. Fracture Classification

The incidence of AVN is presented by fracture type (displaced vs. undisplaced) and according to three different fracture classifications: the Garden classification, Pauwels’ classification, and the Delbet classification (pediatric studies only). The commonly used Garden classification was first described in 1961 [8] and describes fracture severities using staging (I–IV), based on the degree of fracture displacement as observed on anteroposterior radiographs [8,9]. Pauwels’ classification has been in use since 1935 and was the first developed biomechanical classification for femoral neck fractures, although its significance has since somewhat declined [10]. Femoral neck fractures are classified into Pauwels degrees I–III according to the angle of their inclination relative to the horizontal plane and, therefore, the forces acting on the fracture line [10,11]. Delbet’s classification is widely used to describe pediatric hip fractures and was first published in 1907 [12]. Delbet’s classification defines fracture types I–IV based on the anatomic location of the fracture line [12], which is related to the risk of AVN development [13]. The three classifications are described in Table 2.

#### 2.4.2. Patient Age

The effect of age on AVN risk was assessed by analyzing the incidence of AVN in patients aged 20–50 years vs. those aged > 50 years. This age cut-off was selected to assess the possible differential risk of AVN in middle-aged vs. younger patients in light of clinical guidelines recommending arthroplasty or hemiarthroplasty in elderly patients with displaced fractures [14,15].

#### 2.4.3. Meta-Analysis

Due to the apparent heterogeneity of both study designs and their real-world settings, no formal testing for heterogeneity was employed. A random-effects model was used to estimate the pooled incidence of AVN depending on fracture type (displaced vs. undisplaced), Garden stage, Pouwels’ degree, Delbet stage, and age category (20–50 years vs. >50 years). The meta-analysis results are presented as pooled means with a 95% confidence interval (CI).

#### 2.4.4. AVN Incidence and Time Interval from Injury to Surgery

The incidence of AVN in patients undergoing fracture stabilization surgery was assessed with respect to the the time interval between injury and surgery by using a general regression model. Mean or median time from injury to surgery, whichever was reported, was used in the analysis. The maximal reported time interval was used for studies that reported all surgeries within a given number of hours from injury.

## 3. Results

### 3.1. Study Selection

The search strategy rendered 89 citations, of which 22 were excluded during the title and abstract screening stage so that a total of 67 full-text articles were screened for eligibility. Of these, 15 were excluded because the vast majority (*n* = 10) were unavailable in English. Overall, 52 articles were included. Outcomes of the selection process are documented in the PRISMA flow diagram (Figure 1).

### 3.2. Overview of Included Studies

The majority of included studies were retrospective (*n* = 43). The sample size ranged from 17 to 417 participants. Three studies reported solely on pediatric populations [16,17,18], while nine studies included only participants above 50 years of age [19,20,21,22,23,24,25,26,27]. The studies included those that reported the use of both closed and open fracture-reduction techniques and employed a wide range of implants, most commonly cannulated screws (28 studies). Only one study described conservatively managed patients, and no comparison of AVN risk between surgical and non-surgical treatment could be made [18]. An overview of the included studies is presented in Table 3.

AVN was reported across the included studies in 0–53.4% of patients whose fractures were surgically managed. The timeframe from surgery to AVN detection was broad, ranging from as early as three months [28] up to 5 months [21,29,30] or more [31,32] years post-surgery.

Bali et al. was the only study that reported on 13 pediatrics patients managed conservatively in addition to 23 pediatrics patients who received surgery [18]. Although the incidence of AVN after a mean follow-up of 2.7 years (range, 1.1–9.5 years) was 16.7% in patients who received surgical treatment and 2.8% in those managed conservatively, fracture reductions were lost in 8 out of 13 patients, requiring surgical stabilization [18]. Furthermore, patients who were managed conservatively were solely those with undisplaced fractures (*n* = 8) and those in whom other concomitant injuries prohibited surgical management (*n* = 5) [18]; in this case, the comparison of AVN risk between surgically and conservatively managed patients in this study should be interpreted with caution.
ijerph-19-10050-t003_Table 3Table 3Overview of studies included in the review.Study IDCountryStudy Design *Cohort Size, NInterventionPost-Surgery Follow-UpBaseline AgeDisplaced Fractures, N (%)Undisplaced, N (%)Bajada 2015 [19]United KingdomRetrospective study111 fractures in 108 patientsInternal fixation with cannulated screws≥90 daysMean 79 years (range; 60–96 years)NA111, (100%)Bali 2011 [18]IndiaRetrospective study36Closed or open reduction and internal fixation. Implants included partially threaded cancellous screws or pediatric dynamic hip screwMean 2.7 years (range, 1.1–9.5 years)Mean 10 years (range, 3–16 years)28, (77.8%)8, (22.2%)Chen 2017 [33]ChinaRetrospective study86Closed reduction with cannulated compression screw or dynamic hip system bladeMean 27 months (range, 24–36 months)Mean 53.8 years (range, 26–83 years)42, (48.8%)44, (51.2%)Do 2016 [34]NorwayRetrospective cohort study383Fixation with 2 parallel screws or 3 screws Median 77 months (range 23–125 months)Median 81 years (range 72–86 years)NA383, (100%)Duckworth 2011 [35]United KingdomProspective single-arm study122Closed or open reduction and fixation using three cannulated screwsMean 58 months (18–155 months)Mean age 49 years (range, 17–60 years)122, (100%)NAElgeidi 2017 [36]EgyptRetrospective study35Closed reduction and internal fixation using dynamic hip screw and fibular strut graft27.2 months (range 6–41 months)37 years (range 20–50 years)35, (100%)NAFan 2017 [37]ChinaRetrospective study65Closed reduction and internal fixation with 3 screws≤2 yearsMedian age 61 years (range, 19–84 years)26, (40%)39, (60%)Gregersen 2015 [20]DenmarkCase–cohort study322Fixation with 3 cannulated screws2 yearsAll patients ≥ 65 years, mean 82 years (±8.3)155, (48%)167, (52%)Han 2017 [38]KoreaRetrospective study53Closed reduction and internal fixation with 3 cannulated screwsMean 29.5 monthsMean age 59.1 years (range, 31–82 years)14, (26.4%)39, (63.6%)Hoelsbrekken 2012 [39]NorwayRetrospective study337Closed reduction and internal fixation with hip pins>3 monthsMedian age 82 years220, (65.3%)117, (34.7%)Huang 2011 [40]TaiwanRetrospective study146Closed reduction and internal fixation with parallel cannulated screws in inverted triangle or diamond configurationsMean 4.76 years (range, 2–6 years)Mean 46.6 years (range, 17–60 years)42, (28.8%)104, (71.2%)Huang 2020 [41]ChinaRetrospective study67Gotfried closed reduction and internal fixation with cannulated cancellous screwsMean 22.5 ± 11.3 months (range, 11–34 months)All patients ≤65 years, mean 48.9 years30, (44.8%)37, (55.2%)Jo 2016 [42]KoreaRetrospective study 45Closed reduction in displaced fractures and internal fixation with multiple screws or compression hip screw for all fractures≥2 years (range, 24–75 months)Mean 48 years (range 19–69 years)27, (60%)18, (40%)Ju 2016 [16]ChinaRetrospective study58Closed reduction internal fixation (group 1) and open reduction internal fixation (group 2); several types of implants were usedMean 35.1 months (range, 17–61 months)Mean 9.1 years (range, 1 year and 8 months–15 years and 7 months)NRNRJu 2020 [21]ChinaRetrospective study73Closed reduction and internal fixation with cannulated screwsMean 61 months, (range, 13–128 months)Mean 68.22 years (range, 60–85 years)38, (52.1%)35, (47.9%)Kang 2016 [43]KoreaRetrospective study84Internal fixation with cannulated screwsMean 36.8 months (range, 24–148 months)Mean 55.8 years (range, 16–88 years)35, (41.7%)49, (58.3%)Kilian 2018 [44]SlovakiaRetrospective study82Fixation using a locking plate system with telescoping sliding screws (Targon FN implants)Mean 29 ± 7.1 (range, 24–62) monthsMean 71.6 years (range, 30–90 years)40, 48.8%).42, (51.2%)Kim 2014 [22]KoreaRetrospective study58Internal fixation with multiple screwsMean 46.8 months (range; 12–151 months)Mean 77.5 years (range 65–96 years)NA58, (100%)Kim 2021 [45]KoreaRetrospective study58Internal fixation with cannulated screw, compressive hip screw fixation or nailingMean 23.9 months (range, 4–242 months)Mean 40.0 years (range, 9–80 years)41, (70.1%)17, (29.3%)Kumar 2014 [46]IndiaRetrospective study62Closed or open reduction and internal fixation with cannulated screws, or open reduction and fixation with a dynamic hip screw and a derotationscrew≥2 years (range, 2.5–4 years)Mean 57.2 years (range, 45–82 years)38, (61.3%)24, (38.7%)Li 2018 [47]ChinaRetrospective study185Study group (group A, treated with three cannulated screws with DCIABG) Control group (group B, treated with traditional three cannulated screws without DCIABG)Mean 29.26 months in group A and 28.74 months in group BMean 39.1 years in group A and 35.5 years in group B185, (100%)NALiu 2013 [48]ChinaProspective study45Internal fixation with 3 cannulated screwsMean 39.8 monthsMean 45.4 years (±14.2 years)24, (53.3%)20, (44.4%)Luo 2017 [49]ChinaRetrospective study17Open reduction and internal fixation with a modified dynamic hip screw loaded with autologous bone graft≥24 months (range, 24–36 months)Mean 37.2 years (range, 27–52 years)NRNRManohara 2014 [23]SingaporeRetrospective study100Internal fixation with cancellous screwsMean 39 months (range, 25–76 months)Mean 78 years, (range, 61–94)NA100, (100%)Min 2011 [28]KoreaRetrospective study146Open or closed reduction and fixation with a slidinghip screw or cannulated screwsmean 5.2 years (range, 3 months–7 years)mean 45.7 years (range, 17–70 years)91, (62%)55, (38%)Mukka 2020 [24]SwedenProspective pilot study235, including 65 patients treated with internal fixation and of interest to the reviewInternal fixation with two cannulated screwsMedian 26 months (range, 0–56 months)Median 83 years (range 61–98 years) NA65, (100%)Novoa 2019 [29]SpainCase-control study121Internal fixation with cannulated titanium screws with a diameter of 6.5 mmMean 76.2 ± 31.6 months in ANFH group and 52.6 ± 25.1 months in the control group Mean 63.7 years (range,44–83) in the ANFH group and 69.7 years (range, 18–93) in the control groupNA121, (100%)Osarumwense 2015 [50]United KingdomRetrospective study43Closed reduction (displaced fractures only) and fixation with the Targon FN implant≥24 months (range, 24–47 months)Mean 66 years (range, 24–94 years)12, (28%)31, (72%)Park 2015 [25]KoreaRetrospective study19Fixation with Proximal Femoral Nail AntirotationMean 53.3 months (range, 30–72 months)Mean 77 years (range, 71–82 years)NA19, (100%)Park 2021 [51]KoreaRetrospective study55In situ or post-reduction internal fixation with three parallel screws Mean 36.3 (median 29.5, range: 12–85) months and 36.2 (median 26.0, range: 13–120) months for the in situ and reduction groups, respectively Mean 52.6 ± 10.3 year for the in-situ group and 51.3 ± 9.8 years for the reduction groupNA55, (100%)Parker 2013 [52]United KingdomRetrospective study320Fixation with a dynamic locking plate (Targon FN)Mean 2.5 years (range, 2.0–10)Mean 76.0 years (range, 22–103)208, (65%)112, (35%)Pei 2020 [31]ChinaRetrospective study250Closed or open reduction and fixation with 2–3 hollow compression screwsMean 7.5 years (range, 1–15 years)Mean 56.4 ± 6.8 years (range, 18–59 years)142, (56.8%)108, (43.2%)Razik 2012 [53]United KingdomRetrospective study92Fixation using cannulated screws, dynamic hip screws, or a dynamic hip screw with a derotation screwMean 2 yearsMean 44.7 years (range, 11–59), (median age, 50 years)68, (73.9%)24, (26.1%)Riaz 2016 [26]United KingdomRetrospective study251Fixation with cannulated hip screwsNRMean 77 years (range 60–101 years)NA251, (100%)Şahin 2020 [54]TurkeyRetrospective study78Closed reduction and internal fixation, (Dynamic hip screw and antirotation screw—group 1, Cannulated screw—group 2)Mean 18.1 months (range, 12–36 months) for group 1 and 14.2 months (range, 12–25 months) for group 2Group 1- (mean age 45.7 years; range, 19–62 years), group 2—(mean age 41.9 years; range, 17–75 years)34, (43.6%)44, (56.4%)Sales 2012 [55]IranProspective cohort study51Fixation using 3 cancellous screws in a reverse triangle arrangement≤1 yearMean 49.12 ± 16.8 yearsNRNRSamy 2015 [56]EgyptProspective study60Group A—closed reduction and internal fixation with three cannulated screws Group B—addition of PRP to internal fixation12–48 months with a mean of 28 monthsRange 20–45 for both groups; mean age 30 ± 7.8:32 ± 6.4 for group A and 28 ± 8.4 for group B60, (100%)NASchweitzer 2012 [57]ChileRetrospective study29Closed or open reduction and internal fixation with cannulated screwsMedian 28 months (range 24–144 months)Mean 46.45 ± 11.59 years29, (100%)NASjöholm 2019 [58]SwedenRetrospective cohort study417Closed reduction and internal fixationMean 3.4 years (range, 2–14)Median 78 years (range, 50–108 years)NA417, (100%)Su 2011 [59]ChinaRetrospective study25Minimally traumatic reduction with K-wires or Steinman pins and fixation with three cannulated screws Mean 4.6 years (range, 3–5 years)Mean 35 years (range, 19–54 years)25, (100%)NASun 2021 [60]ChinaProspective cohort study75Closed reduction and internal fixation using three parallel FTHCS≥2 yearsMean 48.76 years, (range,18–65 years)48, (64%)27, (36%)Wang 2014 [30]ChinaRetrospective study146Open or closed reduction and internal fixation with 2 or 3 cannulated cancellous screwsMean 52 months (range, 9–84 months)Mean 47.5 years (range, 18–68 years)90, (61.6%)56, (38.4%)Wang 2018 [32]TaiwanSingle-centre retrospective study117Closed reduction and unilateral internal fixation with cannulated screws≥2 years (range, 2–8 years)Mean 55.4 years (range, 50–60 years)69, (59%)48, (41%)Wang 2019 [61]ChinaRetrospective study241Closed reduction and internal fixation>18 monthsMean 53.46 yearsNRNRWei 2021 [62]TaiwanRetrospective study22Fixation with single construct with/without an antirotational screw or dual constructsMedian 12 months (interquartile range, 12–24 months)Mean 45.18 years ± 16.00 yearsNRNRWu 2020 [17]ChinaRetrospective study16 patients, 17 hipsOpen or closed reduction and internal fixation using K-wire pinning or screwMean 23.2 months (range 10–58months)Mean 10.4 years, (range, 1–14 years)16, (94.1%)1, (5.9%)Xiao 2018 [27]ChinaSingle-centre, retrospective study36Closed reduction and fixation with dynamic compression locking systemMean 21.58 ± 5.41 months, (range, 12–29 months)Mean 65.33 ± 9.30 years (range,53–82)27, (75%)9, (25%)Xiong 2019 [63]ChinaRetrospective study46Closed reduction and fixation with multiple cannulated screws (6.5 mm in diameter) in inverted triangle or diamond configurationMean 22.0 months (range, 12–36 months)Mean 50.3 years (range, 19–60 years)46, (100%)NAYe 2017 [64]ChinaRetrospective study28Open reduction and internal fixation using cannulated screws and medial buttress plate fixationMean 13.6 months (range, 12–18 months)Mean 42.1 years (range, 29–57 years)NRNRZahid 2012 [65]IndiaRetrospective study33Closed reduction and fixation using 7.0-mm cannulated cancellous screws and fibular strut graftsMean 2 (range, 1–4) years40–60 years (mean, 52 years)33, (100%)NAZeng 2017 [66]ChinaCase-control study325Closed reduction and internal fixation with cancellous screwsMean 42 months (range 37–46 months)Mean 74 years (range 50–94 years)220, (67.7%)105, (32.3%)Zhuang 2019 [67]ChinaRetrospective study26Open reduction and fixation with anteromedialfemoral neck plate with cannulated screwsMean 18 months (range 12–30 months)Mean 36.5 years (range 19–44 years)26, (100%)NA* In some cases, the study’s design was not explicitly reported, and the study’s type was determined by the reviewer based on available methodology. Abbreviations: ANFH, avascular necrosis of the femoral head; DCIABG, deep circumflex iliac artery-bone grafting; FTHCS, fully threaded headless cannulated screws; NA, not applicable; NR, not reported; PRP, platelet-rich plasma.

### 3.3. Incidence of AVN

#### 3.3.1. AVN Incidence in Displaced and Undisplaced Fractures

A total of 25 studies reported both the number of included patients with displaced fractures and the incidence of AVN in these patients [17,18,21,27,28,30,32,36,37,38,39,40,42,43,44,45,50,52,53,56,59,60,65,66,67], and they were included in the meta-analysis. For undisplaced fractures, the corresponding data were available for analysis from 28 studies [17,18,19,21,22,23,24,26,27,28,29,30,32,34,37,38,39,40,42,43,44,45,50,51,52,53,60,66]. The pooled mean AVN incidence was significantly higher among patients with displaced fractures (20.7%; 95% CI: 12.8–28.5%) than among those with undisplaced fractures (4.7%; 95% CI: 3.4–6.0%).

#### 3.3.2. AVN Incidence by Garden Stage

Data on the incidence of AVN by Garden stage were derived from studies reporting both the number of patients with a given stage and AVN incidence for that stage. A total of 10 studies reported Garden stage I fractures [17,21,30,37,38,42,44,45,51,61,66], 10 studies reported Garden stage II fractures [17,21,27,30,37,38,42,44,45,66], 13 reported Garden stage III [17,21,27,30,36,37,38,42,44,45,56,65,66] fractures, and 12 reported Garden stage IV fractures [17,21,30,36,37,38,42,44,45,56,65,66]. The pooled mean incidence of AVN was 13.6% for stage I (95% CI: 0–28.4%); 12.2% (95% CI: 0–26.2%) for stage II; 17.0% for stage III (95% CI: 6.4–27.6%); and 32.8% for stage IV (95% CI: 11.8–53.8%) with no significant differences between the Garden stages.

#### 3.3.3. AVN Incidence by Pauwels’ Degree

Data on the incidence of AVN by fracture degree according to Pauwel’s classification were derived from studies reporting both the number of patients with a given degree and AVN incidence for that degree. Only two studies reported data for Pauwel’s degrees I and II [21,42]. Data from 5 studies were available for degree III fractures [21,42,49,60,64]. The pooled mean incidence of AVN was 21.8% (95% CI: 0–70.4%) for Pauwel’s degree I fractures, 10.3% (95% CI: 0–25.9%) for degree II fractures, and 5.5% (95% CI: 0–12.4%) for degree III fractures, with no significant differences between different Pauwels’ degrees (Figure 2). However, given the small number of studies contributing data, particularly for degrees I and II, conclusions should be cautiously drawn.

#### 3.3.4. AVN Incidence by Delbet’s Type

Delbet’s classification of fractures is used in pediatrics and was represented by a limited number of studies in this review. Only studies reporting the number of patients with a given fracture type and AVN incidence for that type were included. Although three pediatric studies were identified in the review [16,17,18], Ju et al. [16] did not report the incidence of AVN by fracture type; thus, the study was not included in the meta-analysis. This resulted in two studies contributing data for Delbet’s types I, II, and III [17,18] and single research contributing data for Delbet’s type IV [18] due to the lack of patients with type IV fractures in the study by Wu et al. [17]. Increasing Delbet stages represents decreasing AVN risk. This trend could also be observed in the present review (Figure 2). However, the differences in AVN incidence between Delbet stages were not statistically significant. The results should be interpreted with extreme caution due to the small number of pediatric studies.

#### 3.3.5. AVN Incidence by Age Category

Patients’ age was categorized as 20–50 or >50 years old. Data on the incidence of AVN by age were derived from studies reporting both the number of patients within a given age category and AVN incidence for that category. For patients aged 20–50 years, data from four studies were available for analysis [36,47,56,67], while nine studies [19,20,21,22,23,24,25,26,27] contributed data for patients aged > 50 years. The pooled mean AVN incidence was 7.6% among patients aged > 50 years (95% CI: 2.8–12.3%) and was not significantly different from the incidence in patients aged 20–50 years (7.5%, 95% CI: −4.1–19.1.5%).

### 3.4. The Relationship between AVN Incidence and Time Interval from Injury to Surgery

A total of 28 studies reported on both AVN incidence and the time interval between injury and surgery [16,17,22,25,27,28,29,30,32,33,36,40,41,42,43,44,45,47,49,51,52,54,55,56,57,59,65,67]. There was no significant correlation between AVN incidence weighted by sample size and time interval from injury to surgery, with a slope of weighted linear regression equal to −0.01 (*p* = 0.843, R^2^ = 0.01) (Figure 3).

## 4. Discussion

The present review summarised the risk of AVN following surgery to stabilize femoral neck fractures. In the meta-analysis of AVN incidences by fracture displacement, the incidence of AVN was significantly higher in patients with displaced (corresponding to Garden III–IV) compared with undisplaced (corresponding to Garden I–II) fractures. However, when the four Garden stages were assessed individually, no significant differences in AVN incidence between locations were identified. Similarly, no significant differences in AVN incidence were observed between different Pauwels’ angles and Delbet stages. However, due to the small number of studies using clinical classifications, the results should be interpreted with caution. Regarding age, the incidence of AVN was not significantly different between patients aged 20–50 years and those aged > 50 years; however, numerically lower AVN incidence was observed in the latter group. Finally, AVN incidence did not correlate with the time between injury and surgery.

The findings of this review confirm previous reports of increased AVN risk in displaced compared with undisplaced fractures. A meta-analysis by Xu et al., including 2065 patients from 17 case-control studies, demonstrated that the risk of AVN after internal fixation was 0.4-fold higher in patients with displaced fractures (Garden III–IV) than in patients without displacements (Garden I–II) [68].

Regarding the relationship between AVN incidence and the time interval between surgery and injury, the present study’s results are similar to two previously published meta-analyses [68,69]. In a meta-analysis specifically assessing the effect of injury-to-operation interval on the development of late complications, no significant differences in the risk of AVN were detected when comparing surgery performed within and outside of 6, 12, and 24 h intervals from injury and when comparing AVN risk following surgery conducted within 6 h post-injury with surgery performed > 24 h post-injury [69]. Similarly, Xu et al.’s meta-analysis detected no statistically significant difference in AVN risk based on an injury-to-operation interval of ≤24 h vs. >24 h [68].

The observed lack of increased AVN risk in older patients is also consistent with Xu et al., who reported no significant difference in the risk of AVN between patients aged ≤ 60 years and >60 years [68]. It should be noted that the meta-analysis by Xu et al. used a different age categorization compared to the present study in which patients were classified as ≤50 or >50 years old, and the incidence of AVN was numerically higher in the younger patient group. However, both the European Society of Trauma and Emergency Surgery (ESTES) [14] and American Academy of Orthopaedic Surgeons (AAOS) [15] guidelines recommend arthroplasty or hemiarthroplasty in elderly patients with displaced fractures, and these procedures were excluded from the present review. Therefore, it is likely that patients aged > 50 years included in the present review experienced fractures associated with an improved prognosis than younger patients, which influenced the relative incidence of AVN in these groups. Indeed, among the nine studies reporting only on patients aged > 50 years [19,20,21,22,23,24,25,26,27], six studies included only patients with undisplaced fractures [19,22,23,24,25,26]. In contrast, among four studies reporting only patients aged 20–50 years [36,47,56,67], all included only patients with displaced fractures.

The strengths of this review include a substantial number of studies captured and a broad scope of the review in terms of study settings, fracture types, patient age, and the type of implant used. The real-world nature of the included studies supports the applicability of the review findings to everyday clinical practice. Regarding the limitations of this review, no conclusions could be derived on AVN incidence following the stabilization of different fracture types according to Pauwels’ and Delbet’s classifications due to the small number of studies rendering relevant data. Furthermore, while both the Ficat classification and the Steinberg classification are often used to classify the severity of AVN [4], only one of the included studies reported AVN severity according to the Fi-cat classification [54]. This paucity of data on AVN severity meant that it could not be accounted for in the review and meta-analysis. Another important limitation is the heterogeneity of the reviewed studies, which may be a source for bias and does not allow the analysis of AVN risks in specific populations; therefore, more analyses will be planned in order to deepen the knowledge regarding the possible differences in incidence and severity of AVN in selected age groups and after different types of interventions.

## 5. Conclusions

The risk of AVN following femoral neck fractures was substantial and significantly higher for displaced (Garden III–IV) than undisplaced (Garden I–II) fractures. The time interval from injury to surgery did not correlate with AVN incidence. The review results highlight the substantial long-term risk of AVN, particularly in patients with displaced fractures, and call for prolonged post-surgical follow-up for patients with femoral neck fractures.

## Figures and Tables

**Figure 1 ijerph-19-10050-f001:**
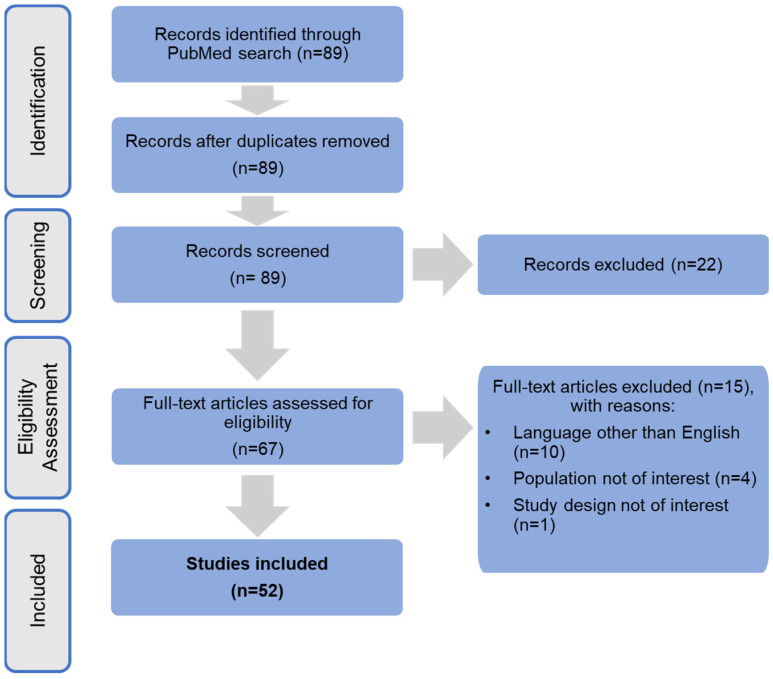
PRISMA flow diagram of the review.

**Figure 2 ijerph-19-10050-f002:**
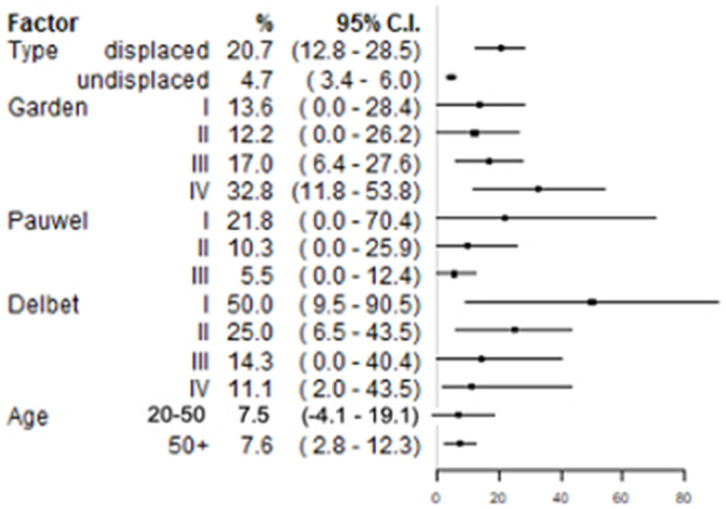
Results of the meta-analysis assessing AVN incidence according to the dependent variables examined.

**Figure 3 ijerph-19-10050-f003:**
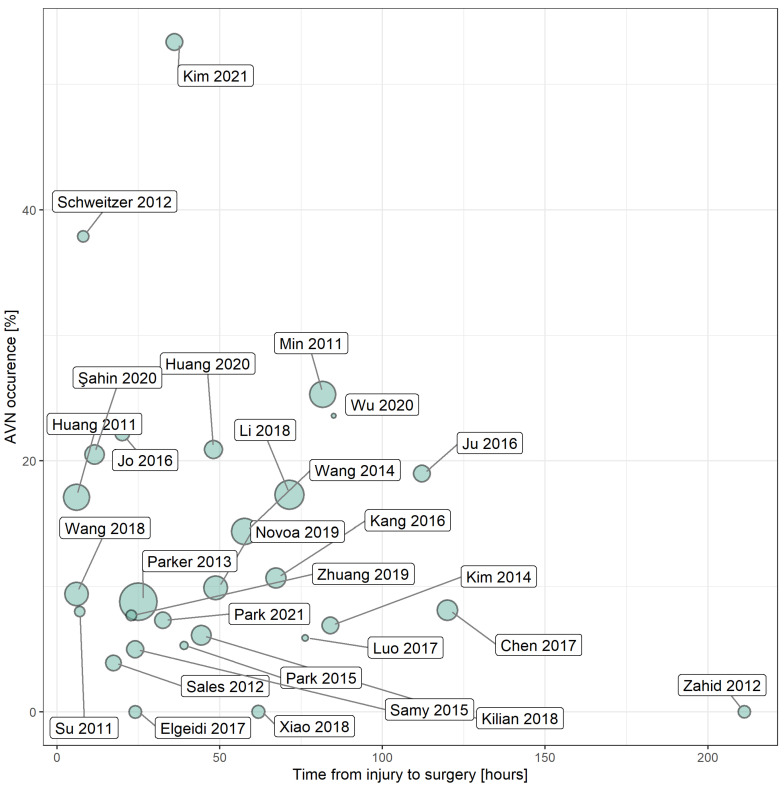
The relationship between AVN incidence and time from injury to surgery [16,17,22,25,27,28,29,30,32,33,36,40,41,42,43,44,45,47,49,51,52,54,55,56,57,59,65,67]. The size of the dot corresponds to the sample size of the study.

**Table 1 ijerph-19-10050-t001:** Inclusion and exclusion criteria.

Studies Included in the Review	Studies Excluded from the Review
Publication date: 1 January 2011 to 22 April 2021 Original research articles Hip-preserving techniques Studies comparing or using different surgical techniques or implants Studies comparing hip-preserving techniques to non-surgical management	Case reports and case series with ≤10 patients Biomechanical, animal studies Reviews and meta-analyses Papers focused primarily on arthroplasty outcomes or patients with neglected hip fractures

**Table 2 ijerph-19-10050-t002:** The Garden [8,9], Pauwels [10,11], and Delbet’s classification [12,13].

Classification	Type	Description	Other
Garden	Stage I	incomplete fracture; undisplaced, valgus impacted	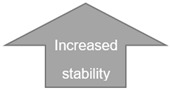
Stage II	complete fracture; undisplaced
Stage III	complete fracture; partially displaced
Stage IV	complete fracture; fully displaced
Pauwel’s	Angle between fracture line and horizontal plane	Dominant forces
Degree I	Up to 30 degrees	Compressive forces predominate
Degree II	30–50 degrees	Shearing stress is present and may adversely impact healing
Degree III	50 degrees and more	Shearing stress dominates
Delbet’s	Type I	Transphyseal fracture, with or without dislocation of the capital femoral epiphysis	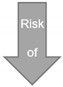
Type II	Transcervical fracture
Type III	Cervicotrochanteric fracture.
Type IV	Intertrochanteric fracture

## Data Availability

Data are available from the corresponding author upon reasonable request.

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
