# Peer review of "The Risk of Avascular Necrosis Following the Stabilization of Femoral Neck Fractures: A Systematic Review and Meta-Analysis"

_ijerph, 2022, doi:10.3390/ijerph191610050_

Round 1

Reviewer 1 Report

See attached file.

Author Response

Dear Reviewer,

Thank you for your interest in our manuscript, comments and suggestions aiming to improve the quality of our manuscript.  We hope that the improvements made will allow our manuscript to meet the requirements for publication in International Journal of Environmental Research and Public Health.

Best regards,

Wojciech Konarski

Reviewer 2 Report

The study is a meta-analysis of the occurrence of AVN after femoral neck fractures. The study is designed and performed well and leads to conclusions that are relevant to the surgical community. An important observation from the data is that there is no correlation between the incidence of AVN and the time until surgery, which is a critical piece of information in the planning of surgery. 

Major comments

1, The rationale for the study is stated in 2.3 as "no overall recommendations on the choice of implant were made". However, the authors do not investigate this question at all. Please use another rationale. 

2, The other key message of the study is the relevance of age in the incidence of AVN. The authors present the full data set in a comprehensive manner and even note that there were a few studies focusing solely on either paediatric or gerontology patient populations, while others have wider age groups. Then the authors choose an arbitrary cut-off age at 50 and analyse the data set as below or over 50 years of age. However, this choice is not supported by the data. For example, including children in the below 50 group is clearly a bias since 10 year olds are not comparable to 45 year olds although they are lumped together. The source studies have three (and not just two) rather distinct age groups: about the ages of 10, 40 or 75, representing clearly different patient populations in respect of healing capacity, bone quality, and injury mechanism. Thus, it is suggested to redo the analysis on these 3 age groups instead of the below / over 50 ages, since these describe the patient population better, i.e. children, middle-aged and elderly. 

Minor comments: 

1, the first sentence in section 2.2. is garbled, please rephrase. 

Author Response

Dear Reviewer

Thank you for your interest in our manuscript,  comments and suggestions aiming to improve the quality of our manuscript.  We hope that the improvements made will allow our manuscript to meet the requirements for publication in International Journal of Environmental Research and Public Health.

The study is a meta-analysis of the occurrence of AVN after femoral neck fractures. The study is designed and performed well and leads to conclusions that are relevant to the surgical community. An important observation from the data is that there is no correlation between the incidence of AVN and the time until surgery, which is a critical piece of information in the planning of surgery.

  • The rationale for the study is stated in 2.3 as "no overall recommendations on the choice of implant were made". However, the authors do not investigate this question at all. Please use another rationale.

We have rephrased this paragraph to clarify the issue. When we designed our review, a Cochrane Review on this topic made no specific recommendations regarding implant type based on all relevant outcomes assessed (such as for example mortality, pain, health-related quality of life, and mobility, in addition to AVN incidence); a result which supported the joint analysis of all implant types we performed.

  • The other key message of the study is the relevance of age in the incidence of AVN. The authors present the full data set in a comprehensive manner and even note that there were a few studies focusing solely on either paediatric or gerontology patient populations, while others have wider age groups. Then the authors choose an arbitrary cut-off age at 50 and analyse the data set as below or over 50 years of age. However, this choice is not supported by the data. For example, including children in the below 50 group is clearly a bias since 10 year olds are not comparable to 45 year olds although they are lumped together. The source studies have three (and not just two) rather distinct age groups: about the ages of 10, 40 or 75, representing clearly different patient populations in respect of healing capacity, bone quality, and injury mechanism. Thus, it is suggested to redo the analysis on these 3 age groups instead of the below / over 50 ages, since these describe the patient population better, i.e. children, middle-aged and elderly.

We agree that more detailed analyses focused on age would be interesting, although these are largely hampered by lack of access to patient-level data. Given the variable reporting of patient age in included publications (many reported mean or median age of patients only with no detailed stratification, although some studies focused solely on specific age groups), the cut-off of 50 years was selected arbitrarily to identify studies specifically focused on the elderly population, in which both European and American guidelines recommend arthroplasty (not captured in the review) as opposed to internal fixation for displaced fractures. Of note, our results indicate that orthopedic surgeons in the real-world largely adhere to these recommendations, as 6 out of 9 reviewed studies focused on the elderly population reported only on patients with undisplaced fractures.

  • the first sentence in section 2.2. is garbled, please rephrase.

Thank you; this sentence has been rephrased.

Reviewer 3 Report

The authors present a study on the risk of avascular necrosis after fixation of femoral neck fractures. It is always interesting to have new information about this highly incident picture. However, the biases that may affect the results of this meta-analysis should be further investigated.

For instance, in the abstract an incidence of up to 53.4% is reported. I consider that it is an outlier that should be investigated and, in any case, should not serve as presentation for the woek.

The quality of the included studies has not been assessed. All are retrospective and many have a small sample size.

It seems that there is a great heterogeneity in the results and the samples that between the different studies, which has not been investigated either.

Author Response

Dear Reviewer,

Thank you for your interest in our manuscript, comments and suggestions aiming to improve the quality of our manuscript. We hope that the improvements made will allow our manuscript to meet the requirements for publication in International Journal of Environmental Research and Public Health.

The authors present a study on the risk of avascular necrosis after fixation of femoral neck fractures. It is always interesting to have new information about this highly incident picture. However, the biases that may affect the results of this meta-analysis should be further investigated.

  • For instance, in the abstract an incidence of up to 53.4% is reported. I consider that it is an outlier that should be investigated and, in any case, should not serve as presentation for the woek.

The relevant fragment of the abstract has been rephrased.

  • The quality of the included studies has not been assessed. All are retrospective and many have a small sample size.

Since our review focused on real-world evidence solely, we have refrained from conducting a formal quality assessment, such as using the Downs and Black checklist. Instead, we have endeavored to describe the included studies comprehensively in Table 3 allowing the reader an insight into the quality of the evidence.  Of note, several prospective studies were included in the review, as stated in Table 3.

  • It seems that there is a great heterogeneity in the results and the samples that between the different studies, which has not been investigated either.

A random effects model was used for all meta-analyses (as stated in 2.4.3). This type of model allows for the possibility that included studies are heterogenous. In the analysis of AVN incidence variability relative to time from injury to surgery, weighting by sample size was applied and we have also visually accounted for sample size in Figure 3.

Round 2

Reviewer 2 Report

Most of my comments were answered and the revised manuscript is significantly improved. However, the rebuttal for grouping pediatric and adults patients in the same group is not acceptable: these are clearly different patient populations and must be analysed as such especially in a review paper. 

Author Response

Dear Reviewer, 

Thank you very much for your comments and thoughts. They are very valuable.

Although we understand your concern, we would like to underline that our analysis aimed at assessing the AVN risk in unselected populations, regardless of age and type of intervention, which in our views reflect better real-world data and can wider support daily clinical practice. This is clearly expressed in our revised manuscript in the study rationale section.

Based on your questions raised in this and previous review we’re planning to prepare new analysis to fill the knowledge gap in the subject of AVN incidence (eg, regarding the age). Your suggestions are really helpful, giving us the insight on what would be most interesting for the readers, thank you.

Taking into account the discrepancy of the research group, the main assumption of the study was to answer the question: does early surgical intervention reduce the risk of AVN, and we focused on this issue. Assuming different surgical techniques, the size (surface area) of the canals in the femoral neck reaching the femoral head for a given type of fixation are similar (for a short gamma intramedullary nail it is 0.9 cm2, for DHS 1.2 cm2, for screws (depending on whether 2 or 3) it is from 0.7 to 1.1 cm2), in adolescents (with Kirschner wires, depending on the number of wires) it is from 0.5 to 1 cm2). Therefore, all data has been collected together. Comparing the surface area of each anastomosis, it degrades roughly the same amount of bone tissue, and thus can impair blood supply in a very comparable manner.

I sincerely hope that the work will be recognized and accepted.

Best regards

Reviewer 3 Report

The changes made are insufficient. Although the authors claim to have applied a broad approach based on studies that approximate the real-world scenario, I still detect important sources of bias and heterogeneity: most of the included studies are of poor quality and include groups of patients and interventions that are very different from each other. I believe that the authors should reconsider to focus the article objectives to increase the validity of their results.

Author Response

Dear Reviewer,

Thank you very much for your comments and thoughts. They are very valuable.

We understand your concerns – the general approach we chose to describe the risk of AVN doesn’t allow for deepening this subject in more specific populations, eg, selected by age and/or type of intervention. At this moment we can only address this in the limitations, but we’re planning to prepare separate, more focused publications.

Taking into account the discrepancy of the research group, the main assumption of the study was to answer the question: does early surgical intervention reduce the risk of AVN, and we focused on this issue. Assuming different surgical techniques, the size (surface area) of the canals in the femoral neck reaching the femoral head for a given type of fixation are similar (for a short gamma intramedullary nail it is 0.9 cm2, for DHS 1.2 cm2, for screws (depending on whether 2 or 3) it is from 0.7 to 1.1 cm2), in adolescents (with Kirschner wires, depending on the number of wires) it is from 0.5 to 1 cm2). Therefore, all data has been collected together. Comparing the surface area of each anastomosis, it degrades roughly the same amount of bone tissue, and thus can impair blood supply in a very comparable manner. Therefore, we believe that the heterogeneity of the research group is not of great importance in our publication.

I sincerely hope that the work will be recognized and accepted.

Best regards,